# Exploring the Role of Family and School as Spaces for 1.5 Generation South Korean’s Adjustment and Identity Negotiation in New Zealand: A Qualitative Study

**DOI:** 10.3390/ijerph17124408

**Published:** 2020-06-19

**Authors:** Su Yeon Roh, Ik Young Chang

**Affiliations:** 1Department of Exercise Rehabilitation and Welfare, College of Health Science, Gachon University, Inchon 21936, Korea; Dr.Rohpilates@gachon.ac.kr; 2Department of Community Sport, Korea National Sport University, Seoul 05541, Korea

**Keywords:** family, school, 1.5 generation, South Korea, New Zealand, adjustment, identity negotiation

## Abstract

To date, the majority of research on migrant identity negotiation and adjustment has primarily focused on adults. However, identity- and adjustment-related issues linked with global migration are not only related to those who have recently arrived, but are also relevant for their subsequent descendants. Consequently, there is increasing recognition by that as a particular group, the “1.5 generation” who were born in their home country but came to new countries in early childhood and were educated there. This research, therefore, investigates 1.5 generation South Koreans’ adjustment and identity status in New Zealand. More specifically, this study explores two vital social spaces—family and school—which play a pivotal role in modulating 1.5 generation’s identity and adjustment in New Zealand. Drawing upon in-depth interviewing with twenty-five 1.5 generation Korean-New Zealanders, this paper reveals that there are two different experiences at home and school; (1) the family is argued to serve as a key space where the South Korean 1.5 generation confirms and retains their ethnic identity through experiences and embodiments of South Korean traditional values, but (2) school is almost the only space where the South Korean 1.5 generation in New Zealand can acquire the cultural tools of mainstream society through interaction with English speaking local peers and adults. Within this space, the South Korean 1.5 generation experiences the transformation of an ethnic sense of identity which is strongly constructed at home via the family. Overall, the paper discusses that 1.5 generation South Koreans experience a complex and contradictory process in negotiating their identity and adjusting into New Zealand through different involvement at home and school.

## 1. Introduction


*South Korean parents are most anxious to ensure their children are well-schooled, spending around $6 billion a year to send them to study abroad in countries like New Zealand-but they still disapprove when their offspring adopt Western ways.*
(New Zealand Herald, 24 June 2008)

According to Hall [1] identity is produced within specific social, political, cultural and historical contexts. Moreover, there is increasing agreement amongst scholars that people have distinct identities at different times and spaces such that identity negotiation is an on-going process [1,2,3]. Although it is certainly not a new phenomenon, within a contemporary and increasingly globalised context the process of migration highlights a wide range of identity-related issues. Migration-related identity discussions are becoming more complex given that immigrants’ lives are deeply embedded in “transnational social fields” that maintain “a wide range of affective and instrumental social relationships that link places of origin and settlement” [4] (p. 27).

To date, the majority of research on migrant identity negotiation in transnational migration context has focused primarily on adults. However, identity issues linked with transnational migration are “not only related to those who have recently arrived, but are also relevant for subsequent descendants” [5] (p. 417). Consequently, there is increasing recognition by sociologists [6,7,8,9,10,11] that as a particular group the “‘1.5 generation’ deserves more attention in migration and transnationalism studies” [12] (p. 63). General speaking, the 1.5 generation can be understood as bilingual, bicultural and hyphenated immigrants who are placed between the first and second generation [13].

The term “1.5 generation” was first used as a colloquial term in the 1970s within the South Korean communities within the U.S., especially in Los Angeles and New York, where the majority of South Koreans settled in the 1970s and 1980s. It was officially accepted as an academic term in a Korean American newspaper, *The Korean Times/Hankook Ilbo*, to describe the South Korean immigrants who were born in their home country but are being educated and came to the U.S. before the age of puberty [8,14]. Unlike their parents (first generation) or their siblings and their offspring who were born in the host country (second generation), the 1.5 generation immigrant’s socio-cultural experiences can be different from their parents and offspring. In other words, they are biologically indifferent to the first generation, but demographically close to the second generation, but culturally somewhere between the first and second generation. For those reasons, Rumbaut and Ima [15] (p. 22) define the 1.5 generation as follows:


*Neither part of the ‘first’ generation of their parents, the responsible adults who were formed in the homeland, who made the fateful decision to leave it and to flee as refugees to an uncertain exile in the United States, and who are thus defined by the consequences of that decision and the need to justify it; nor are these youths part of the ‘second’ generation of children who are born in the USA, and for whom the ‘homeland’ mainly exists as a representation consisting of parental memories and memorabilia, even though their ethnicity may remain well defined.*


What is unique about the 1.5 generation is that their migration-related identities and adjustment can be quite different from those of both their parents and offspring. However, defining who belongs to the 1.5 generation can be much more complicated. Most research conducted on the 1.5 generation cannot clearly state or agree on who they are. As a result of this ambiguity, a number of slightly different definitions and classifications are available, depending on factors such as the period they have been exposed to the cultures of the countries of origin and destination [7,16,17]. Based on the period of education in the host country, Gans [18] categorized the 1.5 generation into three specific groups: (1) 1.25 (ethnic identity: close to the first generation and home country), (2) 1.5 (in-betweenness: function in both societies) and (3) 1.75 generation (local identity: close to the second generation and host country). Although each group revealed a different range of “in-betweenness”, most researchers would agree that in broad terms, the 1.5 generation has one thing in common—that it cannot be classified in the same way as either the first or the second generation. Indeed this generation represents a significant intersection/nexus in identity construction since for individuals it holds the possibility of alienation, accepted neither here nor there.

The concept of in-betweenness in many migration studies has been used to explore non-white 1.5 generation identities in specific situations such as: (1) between adulthood and childhood, (2) between home and host countries, and (3) between majority and minority cultures in a host society [12,19]. However, the concept of in-betweenness in previous research raises a couple of questions: where can we see and experience a sense of in-betweenness? and in what kind of social spaces can in-betweenness of the 1.5 generation be manifested? As Bairner [20] states: “social spaces such as family and school are hugely important in the construction and reproduction of memories and identities” (p. 112). Thus, this research focuses on social spaces where the in-betweenness of 1.5 generation Korean-New Zealanders is represented.

Particularly for the 1.5 generation, family and school contexts are very important spaces for two reasons. First, these spaces are where they spend most of their time. Second, these spaces are where their “in-between” identity can be negotiated through continuous interaction with cultures of the countries of origin and destination. Moreover, Portes and Rumbaut [9] assert that the family context can represent the cultures of both the home country and the first generation while school can symbolise the cultures of the new host country and the second generation. Therefore, the family and school environments play crucial roles as spaces which can affect the 1.5 generation’s adjustment and identity negotiation in transnational and intergenerational context.

Therefore, this study examines the process and experience of transnational migration of the South Korean 1.5 generation living in New Zealand. More specifically, this paper: (a) outlines the-methods used; (b) examines how the migration decision by parents affects 1.5 generation South Koreans’ identity negotiation at the early stage of settlement and explores how 1.5 generation South Korean adjusts into New Zealand and negotiate their identity through the experiences at home and school; (c) discusses the complexities and contradictions of how the South Korean 1.5 generation adjusts into a new country and negotiates their identity between family and school spaces; and (d) offers a conclusion and considerations for future research.

## 2. Methods

### 2.1. Research Design

It is important to recognise the complexity of the lives of 1.5 generation in relation to how they adjust and negotiate their identities in a new country. As Kivisto and Faist [21] (p. 13) point out, global migration has been a pervasive influence on all our lives and has changed political, economic, social and demographic structures in both the countries of origin and destination. To understand complex global migratory processes, this research utilises a qualitative study, which provides unique and valuable information about the 1.5 generation South Koreans’ adjustment and their identity negotiation in/to New Zealand

More specifically, this study utilises the qualitative method of in-depth interviewing. Kvale and Brinkmann [22] describes qualitative interviews as “a construction site of knowledge” (p. 2), where two or more individuals discuss “a theme of mutual interest”. Therefore, interviewing provides a key source of knowledge and understanding of the social reality from individuals and allows for a unique insight into understanding other people’s experiences and the meanings they have constructed through their experiences [23].

### 2.2. Participants

Semi-structured interviews were held with 25 South Korean 1.5 generation in New Zealand in order to gain insights into the construction of their identity, adopting purposive sampling and snowball sampling. All interviewees: (1) migrated to New Zealand with their families, (2) started their New Zealand education in elementary, intermediate or high school and (3) have lived in New Zealand for at least 15 years. The ages of interviewees ranged from 25 to 36. They mainly resided in the cities of Auckland, Christchurch, Hamilton and Dunedin where the (tertiary level) South Korean Student Association is well established and where many South Korean immigrant families have settled. All interviewees have graduated from high schools and universities in New Zealand. As a result, all informants in this study could speak both Korean and English fluently. A description of the participants is available in Table 1.

### 2.3. Data Collection

The data were collected from semi-structured. A comprehensive interview guide is available in Table 2.

Audio-taped interviews were conducted in the participants’ language of preference (Korean, English or both) for 60 to 90 min. Notes were made immediately after the interviews and during transcription and were appended to the transcripts. All interviews were conducted in person at a location of the interviewees’ choice. For interviews conducted in Korean, translations are provided and reviewed by the authors. Additional information was obtained through e-mail when needed in the analysis processes.

### 2.4. Data Analysis

The transcribed interviews were coded for the purpose of organising dominant themes and finding relationships among various major categories emerging from the respondents’ narratives. Through the initial coding process, authors conducted open coding of the data, with the focus on generating categories and seeking to determine how categories vary.

The data interpretation follows four phases: describing, organising, corroborating/legitimating, and representing the account. Describing is a reflective process in which the analyst steps back from the field and reviews what authors grasps from the given data. Analysis continues with organising proper themes and subsequently finding theoretical units present in the data. In the organisation of themes phase, researcher explores relationships among the features described by participants in the research field. This phase is called the corroborating/legitimating process of reviewing the data. Finally, the interpretive process is completed when the themes are compared and contrasted with other data sets in order to gain a better understanding of the data.

## 3. Results

### 3.1. Almost Everything Was Not My Decision: Early Adjustment in New Zealand

There are differences in roles between parents (first generation) and their children (1.5 generation) in the processes of migration decision making [7,8,24,25]. The main dissimilarity is the different degrees of authority on the migration decisions. While the migration decision is seen as the outcome of family members’ consensus, children not only have an unequal stake in the migration decision-making processes, but also are almost inevitably in an invisible role vis-à-vis their parents [26]. According to Bartley [25] (p. 386), children as dependents “may be consulted in the matter, but rarely are they in a position to decide for themselves to migrate, nor are they necessarily able to process the long-term implications that such a decision is likely to have on their lives”.

In the context of lifestyle migration, many South Korean immigrants have decided to migrate to New Zealand in search of a better quality of life and education for children, but children’s opinions in the processes of the migration decision have been mostly ignored by parents because parents believed that their children were too young to discuss migration. According to Bartley and Spoonley [12] (p. 74), “the reason for their lack of involvement in the decision may have been their age, as many of the participants were still young children when the decision was made”. Therefore, immigrant children not only put under a considerable amount of pressure to migrate without any information, but also be less involved in or totally excluded from access to mobility against their own will.

Interviewees in the present study stated that while some were aware of their family’s migration to New Zealand, others did not get sufficient opportunities to discuss on the matter with their parents. Some even confessed that they did not know where they were moving to until the very moment they left South Korea. Therefore, most South Korean immigrant children were not prepared for migration, leaving them to be puzzled about the nature of their migration and how they were supposed to adapt to their new surroundings.


*I did not exactly know when my parents decided to migrate to New Zealand. One day they told me, we are going to New Zealand and then after two weeks, we moved. They prepared for migration over a long time, but they did not say anything to us. (At that time) I did not know both where New Zealand was located and what kind of language they (New Zealanders) spoke.*
(Personal interview with L.H.P.)

As the result of the lack of preparation and information regarding migration, the cultural transition into a new country was not smooth at the early stage of settlement. Some interviewees who confessed that they were less prepared for migration were confronted with some incidents which may have affected their identity negotiation in a new country. In particular, they felt frustrated when having a communication problem in English with native speakers in various settings including the corner dairy to general English at school. Due to these experiences their cultural transition into New Zealand was affected in a negative way, unwilling to be assimilated into the new country.


*So, I turned five and started going to school and I knew no English, so I couldn’t speak anything. So the first day of school, I sat outside the class and I cried because I was (totally) lost. And then, a Kiwi lady came up and gave me two dollars and told me something that I didn’t understand at all and I went into class and I stayed in class not understanding anything. And then I met a few Korean people around my house who went to the same school around my age so I started hanging out with them, mainly because I couldn’t speak anything. So I did that. And, I got along with them ok but a few times, I got in trouble at school during primary years.*
(Personal interview with P.S.J.)

Most of the interviewees stated that they were already immersed and socialised into their home cultures as they spent most of their childhood in South Korea. In other words, they described themselves as Koreans at the early stage of settlement because of long experiences in their home country. As a result, after migration, 1.5 generation South Koreans did not have difficulties in negotiating ethnic identity, but they experienced difficulties in adjusting to the host country like making local friends and improving English skills:


*I think my age was one of the reasons. I almost finished 3rd year of middle school in South Korea (year 10 in NZ) before I came here, so I felt more comfortable with Koreans, and I did not even think of having foreign (Kiwi) friends. I had some Asian, Maori and Samoan friends but had never met them outside school. So it resulted in less chance of using English.*
(Personal interview with P.B.H.)

According to Portes and Rumbaut [9], where immigrant parents and children were born and how long they spent in their birthplace affect the definition of their ethnic identities. The findings point out that 69% of foreign-born children (1.5 generation) who are accompanied by their parents define themselves as their home country’s nationality, while only 31% saw themselves as a hyphenated nationality. If both parents were born in the same country, their children (about 90%) were “much more likely to that national origin as part of their own identity” [9] (p. 166). In the same vein, Ip’s study [27] on rethinking contemporary Chinese circulatory transmigration: a New Zealand case shows the similar result. Of the Chinese immigrants who have migrated to New Zealand since 1987, when New Zealand opened its door to all immigrants regardless of race, national origin and religion, only 5.5% claimed their identity as a hyphened identity (Chinese-New Zealander), while over 94.5% of them identified themselves as their home country identity such as Chinese, Taiwanese or Hong Konger [27]. To sum up, it is important to note that experiences of socialisation in the home country play a significant role in defining immigrant children’s identity in a new country of destination. Therefore, 1.5 generation South Koreans who migrated to New Zealand relatively late felt much closer to South Koreans living in their home country in the issue of ethnic and cultural identities.

### 3.2. Different Identity Negotiation in Different Spaces: Family and School

This section reveals how the South Korean 1.5 generation negotiates their “in-between” identity differently within the home and school context. More specifically, I examined the process of the South Korean 1.5 generation’s identity negotiation by comparing and contrasting the emphasis on traditional cultures at home with that of the school and related social environments.

#### 3.2.1. Family as a Transmitter of the Home Country’s Culture

Many of the interviewees stated that they were predominantly immersed into New Zealand culture through the educational system. However, they also retained their Korean heritage through the family where the values and norms of their parents often prevailed. Most interviewees in this study stated that their family environment was maintained as Korean after migration and the South Korean 1.5 generation took this for granted;


*We just eat Korean food, watch Korean dramas, and just read Korean books mainly (at home). But for the first couple of years, I just read English books and things to improve but after that, my dad was just always sending Korean books and I was just reading Korean news internet stuff. And, it was very Korean. But, it was very natural and comfortable for me as well. That’s what I did.*
(Personal interview with S.S.)

Phinney et al. [28] assert that family is a crucial contributor in negotiating immigrant children and adolescents’ identity, especially in the context of migration. In particular, many Asian immigrant parents moving to Western countries believe that maintaining traditional collectivist family values such as close relationships and fulfilling obligations is even more important while residing abroad. One interviewee stated that “after migration, my father always emphasised family is the number one priority. You should take care of your family first and then your work” (Email interview with C.E.C.).

Almost all interviewees also acknowledged that their parents expected children to retain traditional values such as obedience to parents, obligation to family, politeness to others and respect for the old and such traditions were ingrained in South Korean 1.5 generation children. In this way, immigrant children realised that their Korean attitudes were positively perceived and helped them to be recognised as polite and courteous people in New Zealand society. Consider the quotes of one interviewee:


*My dad always emphasised “be polite as much as you can, especially to old people”. I think the advice from my dad was ingrained in my attitudes. So I behave in that way to elderly Koreans as well as Kiwi South Koreans and even Kiwis recognise me as a really polite young man. For example, when I work, most of my patients are Kiwis and they like my polite attitude and Korean ways.*
(Personal interview with O.K.)

However, there has been a long-standing “intergenerational conflict” between the first generation and their children. Intergenerational conflict within immigrant families emerges from exposure to the rapid cultural adjustment of children to their new country, as compared to that of their parents [11]. In other words, while many immigrant parents struggle with difficulties with language deficiencies leading to unemployment and underemployment, their children rapidly acquire language and adjust to the culture of their new host country.

Bartley and Spoonley [12] give us a good example of Asian immigrants’ difficulties with adjusting to a new country. Many East Asian immigrants in New Zealand encounter unfavourable situations, “where they struggle with the language, their qualifications are often unrecognised by professional bodies such as the New Zealand Medical Association and they are shunned by employers for lacking New Zealand experience” [12] (p. 75). Under such circumstances, parents are heavily dependent on their children who play a role “as a cultural interpreter in dealing with social institutions (school, hospitals and social services) and the host society’s culture” [29] (p. 79). Some scholars [9,11,30] define this phenomenon as the “role reversal” which refers to the fact that parents are now dependent on their children. This role reversal was confirmed during interviews with the South Korean 1.5 generation;


*When I was in high school, my dad often asked me to call his business partners for him. But I felt a lot of pressure because if I make a mistake on the phone, my dad’s business would lose lot of money. Before I was on the phone, I only had a (written) scenario my dad gave me without an accurate understanding about his business. Sometimes, when they (business partners) had questions which were not within the scenario or which I did not understand and when I did not give satisfactory answers to the questions, I was so frustrated. The same situations happened routinely and caused me a great deal of stress. Paradoxically, however, that (calling to other people for my dad) helped me to improve my English on the phone, but to be honest, I was really scared.*
(Personal interview with C.S.W.)

One interesting finding was that under such stressful conditions, many South Korean immigrants, including both the first and 1.5 generation, stated that after migration, they have spent more time with family, doing leisure activities such as sports which played a key role in maintaining strong family bonds. One interviewee described the role of sport in maintaining strong family bonds after moving to New Zealand;


*We (my parents and I) would go out and play golf every now and then. I think sports had a big impact in bonding our family together. Through the time we spent together through sports activities, I think the importance of family was built in my mind naturally.*
(Email interview with O.K.)

More than two thirds of the interviewees shared the view that if their family had not migrated to New Zealand, their parents would have worked very hard all day and children would have to study very hard resulting in minimal quality family time. They stated that after migrating to New Zealand, their parents tried to spend more time with them, playing sports and going on family trips which they would not be able to do in Korea.

Therefore, for the South Korean 1.5 generation family plays a significant role in strengthening intergenerational ties between the first and 1.5 generations but also in retaining South Korean cultural values. As a cultural transmitter, family contributes to negotiating their identity of origin country.

#### 3.2.2. Negotiating a Transnational Identity in School

As noted in the previous section, South Korean immigrant families socialised their children at home with traditional cultural values. However, their children were also exposed to New Zealand culture through influential social environments such as school. According to Portes and Rumbaut [9], school is a space where immigrant children spend more time than any other setting outside their homes and as a result, plays a primary role in acquiring cultures of mainstream society through interaction with English speaking peers and adults.

According to my interviews, school was almost the only space where interaction with local peers was possible. The continual interaction with local peers led to a change in the South Korean 1.5 generation’s sense of identity which was otherwise constructed by socialising with family. Phinney et al. [28] argue that attitudes about integration depend on the amount of time 1.5 generation immigrants have been in their host country. Those who came to New Zealand during primary or early intermediate school made more local friends and kept closer relationships with them over a longer period; on the other hand, as a result of this close interaction with locals and long exposure to New Zealand culture, they gradually ‘lost’ their sense of Korean identity.


*I think it is easy for immigrant students like me to make friends in intermediate or primary schools because most Kiwis who are in primary and intermediate seem not to have any prejudice against races (colours).*
(Personal interview with K.D.H.)


*When I was in inter (mediate), there was no Korean in my class. I did not have a chance to meet and make Korean friends at all at that time. As a result, I naturally became familiar with (local) Kiwi classmates. Even after entering high school, I spent more time with (local) Kiwis and developed friendly relations with them.*
(Personal interview with L.J.Y.)

Interviewees who came to New Zealand relatively late during high school tried to adjust to the new school environment because, as stated earlier, the South Korean 1.5 generation recognised the importance of their parents’ expectations and aspirations for academic achievements. However, they reported having less opportunity to interact with locals due to a short period of residence in New Zealand.


*I came here when I was 15 years old, so I felt more comfortable with Koreans. I did not even think of having local (Kiwi) friends. I had some Asian and other coloured friends, but I had never met them outside of school.*
(Email interview with P.B.H.)

In contrast to the school setting, the interaction between the South Korean 1.5 generation and locals was reinforced in the cultural context such as sport, regardless of the duration of residence. Interviewees seemed to believe that participating in extracurricular activities such as sporting clubs in school could not only increase their friendship with local peers, but also place them closer to mainstream society. One interviewee reported that when he joined the school football team, he felt free from ethnic and racial identity issues because his local peers judged him based on his motor skills.


*Yes, I had a lot of football friends. I didn’t really have Koreans in my school, so, they were all Kiwi but it didn’t matter. We played football every lunchtime, but they judged me by my skill, not my colour.*
(Personal interview with S.S.)

Therefore, participation in extracurricular activities in school played an important role in negotiating the South Korean 1.5 generation’s identity transition from Korean to a hyphenated “Korean-New Zealander” sense of local identity.

Despite the positive influences extracurricular activities in school have on the negotiation of the South Korean 1.5 generation’s identity, they faced a variety of problems within school. Those interviewed identified two main factors that inhibited their full adjustment into New Zealand; (1) “differentiation” in transnational contexts between South Korea and New Zealand, and (2) perceived and experienced discrimination. Firstly, “differentiation” resulting from cultural and physical differences hindered the forging of strong relationships with peers. As a result, many of the South Korean 1.5 generation argue that there was a cultural and emotional “glass wall”, separating them from local peers which they could not overcome. Many interviewees repeatedly spoke of the “absence of *jeong*” which is defined as “a bond of affection or feelings of empathy to others” [31] (p. 285) and “a fundamental substructure of South Korean’s characteristics and the basic psychosocial grammar of South Korean people’s human relationships” [32] (p. 176).


*When I was in the First 11 (soccer), I was the only Korean and Asian in the team. The relations with other members was very good, but it was difficult to open my heart to them because I could not feel a particular feeling of affection for and caring about them (Kiwi friends) although we spent a lot of time together. (There was) No jeong (in a Korean way) between us.*
(Personal interview with P.J.)

In some cases the perceived and experienced discrimination felt by the South Korean 1.5 generation in schools aroused mistrust. In New Zealand where white European culture still dominates, school is a place where stereotypes about Asians, including Koreans persist including: they are good at mathematics, lack English skills and have little involvement or interest in school activities like sports. In most cases, the discrimination perceived and experienced by the South Korean 1.5 generation schools had a negative effect on their adjustment to school and identity negotiation and caused them to rethink their identity. As a result, over time the relationship with local friends often faded or became estranged. Thus, the South Korean 1.5 generation who shared similar experiences with other New Zealanders in their youth eventually formed closer relations with other Koreans and Korean-New Zealanders a point which challenges theories which espouse the natural assimilation qualities of sport.

The history of South Korean immigration indicates a rapid inflow to New Zealand between the mid-1990s to early 2000s. Many first-generation South Koreans expected New Zealand’s education environment to be better for their children, often describing the South Korean education environment as harsh [33]. Despite aspirations to give their children a less competitive education, many South Korean parents still wanted to enrol their children in the most prestigious schools in New Zealand. South Korean parents believed that active involvement in education can help their children to easily adapt school environment and achieve greater academic performance [34]. Consequently, many of the South Korean 1.5 generation were concentrated in school in particular geographic areas. The situation in which the number of South Korean students increased in certain areas meant that the 1.5 generation was exposed to an environment where they were able to socialise more with Korean culture and values, making the opportunity to re-negotiate their identity as South Korean from hyphenated Korean-New Zealander possible. During this process, South Korean 1.5 generation students often established ethnic-based sports and social clubs in schools often competing in friendly games with other ethnic groups from other school. One interviewee who felt he was lacking a Korean identity due to close relations with local Kiwi friends commented on how the ethnic football club affected his identity negotiation:


*A South Korean friend of mind from school asked me to join their team of all Koreans and to play against a Korean team in another school. I said okay but I felt awkward because for a long time, I did not hang out with Koreans but only with Kiwis. However, as time went by, I felt more comfortable with them because we shared lot things like K-pop, K-drama and Korean values.*
(Personal interview with S.S.C.)

This finding confirms Trouille’s [35] observation that participation in ethnic-based sports clubs and competitions may lead to closer contact among immigrant groups and dramatically reinforce their ethnic identity. In other words, an extracurricular activity like football can serve as a space for immigrants to engage in inter-ethnic rivalries, in order to expand ethnic networks and to assert their ethnic identity. Consequently, extracurricular activities in school also play an important role in the formation of the South Korean 1.5 generation’s ethnic identity.

## 4. Discussion

Numerous scholars have documented how first generation, migration decision makers, adapts into a new destination [36,37,38]. In contrast, little is known about how later-generation immigrants negotiate their identities in their new home. Therefore, in this study, I explored identity negotiation in the context of transnational migration by examining how 1.5 generation South Korean-New Zealanders build their ‘in-between’ identity in New Zealand with a particular focus on two key social spaces: family and school.

In the experience of early adjustment in New Zealand, the values and norms of their parents’ culture of origin generally prevail and is a very important agent for maintaining traditional heritage. As Schwartz, Zamboanga, Rodriguez and Wang [39] (p. 161) point out, “parents and family members pass down their ethnic heritage by teaching their children about and exposing them to history, traditions, symbols, historical figures and community member from the family’s heritage culture”. Although imposing the values of parents’ culture of origin helps to maintain 1.5 generation South Koreans’ ethic identity, strict parenting style may have a negative impact on their psychological adjustment in a new country [40,41]. As a result, family in the context of transnational migration was also the primary site of intergenerational conflict, emerging from exposure to the rapid cultural adjustment of children to their new country, as compared to that of their parents [11]. As a result, both generations experienced the “role reversal” [11,30], “with parents suddenly dependent on adolescent (or pre-adolescent) children” [25] (p. 85).

Under such dynamic circumstances in immigrant families, first generation South Koreans tried to become more involved in an active lifestyle and they used various activities such as sports and leisure as a key tool to rebuild a strong family bond. South Korean parents tried to spend more time with their children, playing sports, watching sports games and going on family trips which they could not do in their country of origin. The 1.5 generation agreed with the view that those activities strengthen the family bond. However, some interviews believed that doing something with family members plays an important role in reproducing and imposing Korean traditional values and cultures. Indeed, while the first generation considered spending time with family as a tool to restore family relationships, the 1.5 generation had both positive and negative views on spending time with family. However, both generations did not mention its role in helping their adaptation to new cultures and institutions of the country of destination.

1.5 generation South Koreans were exposed to New Zealand culture through influential social environments such as school. Indeed, school is a space where immigrant children spend more time than any other setting outside their homes and as a result plays a primary role in acquiring cultures of mainstream society [42,43,44]. Therefore, school circumstances both in and out of classroom would lead to the renegotiation of South Koreans’ sense of identity which was mainly constructed by socialising with family and ethnic communities. Such (re)negotiation of identity from ethnic to local is more clearly understood when they join a variety of clubs at school. Interviewees seemed to believe that participating in school clubs not only increased their sense of kinship towards their “Kiwi peers”, but also put them closer to the mainstream society. When they had opportunities to do extracurricular activities with Kiwis in school, they felt they can be freer from national and racial identity issues because their local peers only cared about their achievement regardless of ethnicity. Therefore, participation in club activities played an important role in transforming the identity of 1.5 generation from Korean to “hyphened-Korean” or local identity.

However, 1.5 generation South Koreans faced a variety of problems which may have affected their identity (re)negotiation within the context of schools over time. There are two main causes: (1) the “differentiation” in transnational contexts between South Korea and New Zealand and (2) experienced and perceived discrimination in local context between the mainstream and minority society. Both factors have a predominantly negative impact on their identity negotiation and gave them a chance to look back on their “hyphened identity”. Moreover, despite the aspiration to give their children a less competitive education environment, many Korean parents at this period still wanted to enrol their children in the most prestigious schools in a new country. As a result, South Korean enclaves in school were also built and in turn, they voluntarily established ethnic clubs which played a significant role in strengthening “Korean identity”. Similarly, some scholars [45,46,47] argue that leisure participation not only provides migrants with a sense of belonging, support and social interactions, but also leads them to form ethnic networks and bonds. Therefore, strong ethnic ties can be facilitated by participating in clubs with a similar ethnicity in the context of school.

In summary, 1.5 generation South Koreans’ experiences with their family members and both ethnic and local peers in home and school were interrelated and affected their identity negotiation in the migration context. For 1.5 generation South Koreans interviewed in this research, they seemed to construct ethnic oriented identity which identifies strongly with ethnicity and weakly with dominant culture. However, rather than fixed in a singular identity, they continuously insisted that they can negotiate and perform Korean, Korean-Kiwi or local identity showing flexibility in the different and interrelated situations. In the same vein, Awokoya [42] (p. 277) argues:


*Depending on which identity they aim to acquire or group with which they desire to affiliate, these youth (1.5 and second-generation Nigerians) learn and adopt different social behaviours and learn to engage with others differently in multiple contexts. In these ways, (African) immigrant youth enact their identities and their desires in order to feel a sense of belonging and to be accepted by those with whom they wish to affiliate.*


Within the multiple and interrelated contexts, therefore, 1.5 generation South Koreans experienced a complex and contradictory process in negotiating their identity at home and school.

## 5. Conclusions

This article explored the process of how the South Korean 1.5 generation adjusts into New Zealand and negotiates their identity in two different spaces: family and school which play a significant role as a cultural site and practice.

Both in the process of migration decisions and in the early stage of settlement into the new country, 1.5 generation South Koreans seemed to comply with their parents’ decision to migrate without any complaints, but they struggled to adapt with the new culture of the host country due to: (1) the lack of preparation for migration and (2) experiences of socialisation during their childhood in South Korea. As a result, 1.5 generation South Koreans have a great sense of affirmation and belonging to their ethnic heritage and overall ethnic identity in the beginning of migration into New Zealand.

The family is argued to serve as a key space where the South Korean 1.5 generation confirms and retains their ethnic identity through experiences and embodiments of South Korean traditional values. In this context, although migration from South Korea to New Zealand does affect the South Korean 1.5 generation’s identity negotiation, this research argues that in a new country the family plays a significant role in sustaining the culture, legitimising traditional values and representing “Korean-ness”.

However, school is almost the only space where the South Korean 1.5 generation in New Zealand can acquire the cultural tools of mainstream society through interaction with English speaking peers and adults. Within this space, the South Korean 1.5 generation experiences the transformation of an ethnic sense of identity which is strongly constructed at home via the family. However, they are confronted with a couple of major challenges which affected their identity (re)negotiation: (1) physical, cultural and emotional differentiation and (2) perceived and experienced discrimination in the local context. Both factors have a predominantly negative impact on the transformation of the South Korean 1.5 generation’s identity.

This study indicated that there were two different influential groups—school peers and parents—which influenced 1.5 generation South Koreans’ identity negotiation and adjustment into New Zealand. In school, despite the fact that as many informants insisted, they felt free from ethnic and racial identity issues because their local peers judged them based on their skills and achievement, New Zealand kids may see South Koreans as the “minority” or “the other” based on physical appearance. Moreover, they may hold stereotypes about Asians such as their lack of interest or involvement in sports and other physical activities. However, at the same time South Korean parents (in conjunction with the local South Korean community) are also engaged in a process of Othering. They contrast and critique New Zealand culture by continually reinforcing traditional culture and values. Therefore, future studies are encouraged to explore the nature of this dual process of Othering as part of the migration process within the contexts of cultural activities that may influence identity negotiation and adjustment in a new country. Such analysis will provide a better understanding of how 1.5 generation identity is negotiated through interacting with two different groups engaged in the process of Othering across a range of cultural spaces and practices.

## Figures and Tables

**Table 1 ijerph-17-04408-t001:** Description of the participants.

Name (Initials)	Arrival Date	Age When Migrated in NZ	Nationality	Occupation
J.W.S.	1999	9	NZ	Engineer
C.M.	1993	5	NZ	Lawyer
P.J.	1993	10	NZ	Medical doctor
S.S.	2001	11	NZ	Post-doctoral researcher
O.K.	2000	13	NZ	Dentist
C.S.W.	2001	14	Korean	Restaurant employee
L.J.W.	1999	14	NZ	High school teacher
P.B.H.	2001	15	Korean	Business manager
D.S.J.	2001	13	NZ	Accountant
J.J.S.	2003	10	NZ	Builder
K.J.W.	2002	10	NZ	Chef
H.C.	2002	12	NZ	Account in Korea
D.S.H.	2002	13	NZ	Dental Technician
C.J.K.	2002	10	Korean	Lawyer
K.D.H.	2001	13	NZ	Pharmacist
S.H.J.	2002	12	Korean	Property manager
L.J.Y.	1996	13	Korean	Business manager
S.S.W.	1996	12	NZ	Film producer in Korea
L.J.G.	2003	10	Korean	Business manager
C.A.N.	2000	12	Korean	Plumber
C.E.C.	2000	15	NZ	Chef
L.H.P.	1996	8	NZ	Post-doctoral researcher
S.S.C.	1996	11	NZ	Medical doctor
P.S.J.	2002	5	Korean	Dental technician
C.S.H.	2002	10	NZ	Engineer

**Table 2 ijerph-17-04408-t002:** Interview guide for exploring the role of family and school as spaces for 1.5 generation South Korean-New Zealanders’ adjustment and identity negotiation.

Question	Specific Questions
Personal background	Immigration Status (Nationality) Arrival date in New Zealand (Age) Family relationships Educational (Occupational) background
The involvement in migration decisions	Feeling when your parents decided to migrate to New Zealand Your role among family members in the process of migration decision making to migrate to New Zealand
Early experiences after migration	Your first impressions of New Zealand Perceptions of life in New Zealand (what is it like to be immigrants or from a non-English speaking background, priorities for needs) The cultural differences between South Korea and New Zealand The cultural differences between South Korea and New Zealand and its effect on your identity
Daily life at home after migration	Your experiences at home The big issues in your family Relationship between you and your parents Ways to solve conflicts in your family
School life after migration	Your experiences at school Relationship between you and your local peers or teachers Any conflicts with locals and ways to solve conflicts Conflicts and Solutions and its effect on your identity
A variety of activities at home and school after migration	The main differences in participating in activities like sport and leisure after migrating to New Zealand The role of activities in your life in New Zealand Involvement in a variety of activities (at home and school) Are you involved in any formal clubs in New Zealand and if so which ones and who do you play with (Koreans, Kiwi or mixed)? The main constraints and facilitators when you are participating in activities like sport and leisure at home and school (Such as time and space for activities) The participation in activities and its effect on your adjustment into New Zealand and your identity

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
