# Peer review of "Exploring the Role of Family and School as Spaces for 1.5 Generation South Korean’s Adjustment and Identity Negotiation in New Zealand: A Qualitative Study"

_ijerph, 2020, doi:10.3390/ijerph17124408_

Round 1

Reviewer 1 Report

I enjoyed reviewing this manuscript, titled “Exploring the Role of Family and School as Spaces for 1.5 Generation South Korean’s Adjustment and Identity Negotiation in New Zealand: A Qualitative”.

Thank you for the opportunity to read the manuscript. The New Zealand setting and globalized and transnational context of the migration process by exploring the role of family and school and their identities in the spaces are unique and your research has the potential to provide a useful contribution to the field.

However, in my view, there remain a few important aspects of the paper that require minor revision before it should be accepted for publication. First, I believe that the author may consider changing the research title. It is ‘Exploring the process of South Korean 1.5 generation’s adjustment into New Zealand and their identity negotiation in family and school’, which is more to be in line with findings and more coherent than it is at present. Also, the flow should be improved across the paper. Second, missing data sources in the results section is another concern. To address this concern, I believe that the author should elaborate on how the data were used for analysis in the methods section. Lastly, please review the whole manuscript for grammar (e.g., line 60, remove ‘:’, line 155, ‘others’, not ‘other’ and remove ‘on’,  line 156, remove ‘to’, line 197, remove ‘a’ , line 324, it should be ‘aspirations’, etc….)

Abstract

- The abstract should be more concise to the point.

- Family and School or Home and School, make the terms clearer, not consistent over the paper.

Introduction

- line 11-75 Improve the flow of this introduction and re-arrange contents based on the research title that will be changed.

- line 76-83, the paper about adding details seems like overkill.

Methods

- line 109, Add ‘Migrated Age’ column in Table 1.

- line 110, How the author has collected data through email should have been described.

- line 114, ‘Specific questions’ in table 2 should be adjusted to the left.

- line 116, Add how the author utilized the ‘email interview method’ to collect data

- line 121, Data analysis procedure should be more specific. There was no mention of the literature of data analysis including ‘personal interviews’ and ‘email interviews’ with participants. Add citations in texts and what literature the author referred to.

- line 122, While 25 participants were involved in this study, 10 participants’ data were included in the results section. Please describe the criteria for selecting statements included in results and how to analyze the data.

- line 124, ‘the author or authors’, not ‘I’

Results

- The statements based on interview data from the other 15 research participants, excepting for LHP, PSJ, PBH, SS, Cha, OK, CSW, KDH, LJY, PJ, and SSC, are missing in the results section. Please clarify.

- line 164, ‘(Personal interview with LHP, migrated at the age of eight)’ should be ‘(Personal interview with LHP)’

- line 180, It should be (Personal interview with PSJ). Remove ‘, migrated at the age of five)’

- line 192, ‘(Personal interview with PBH, migrated at the age of fifteen)’ should be ‘(Personal interview with PBH)’

- line 217, ‘3.1. Different Spaces….’ should be ‘3.2. Different Identity Negotiation in Different Spaces

- line 232, Remove ‘, migrated at the age of eleven)’

- line 240, Cha is not included in table 1. Please make it clearer.

- line 251, It will be replaced with ‘(Personal interview with OK)’

- line 276, Same as a previous one, Indicate like this (Personal interview with CSW)

- line 286, It should be ‘(Email interview with OK)’

- line 315, It should be ‘(Personal interview with KDH)’

- line 320, It should be ‘(Personal interview with LJY)’

- line 329, It should be ‘(Email interview with PBH)’

- line 340, It should be ‘(Personal interview with SS)’

- line 358, It should be ‘(Personal interview with PJ)’

- line 390, It should be ‘(Personal interview with SSC)’

Discussion

- line 398, Consider the flow to be related to findings directly

Conclusion

- The paragraphs on line 467-505 feel repetitive of earlier arguments. Repetitive and overlapping arguments and meanings should be removed and rephrased.

- line 492, the term ‘powerful’ seems like subjective expression. Be careful about using the term.

Author Response

Dear,

Thank you very much for your time and effort.

Best Regards, 

Reviewer 2 Report

Review some formatting issues. For example: before the interview appointments appear :/;

I find the work very interesting, although I think I should go into more detail on the interview analyses.

My suggestion would be to analyse gender (woman/man) issues, if there are differences between girls and boys of generation 1.5.

For further work it would be important to analyse issues relating to the transmission of the family language and the difficulties of accommodating the language of instruction in the school.

Author Response

Dear,

Thank you very much for your time and efforts.

Best Regards,
